# Personalized lead exposure information and preventive behaviors in Ivory Coast: Insights from a pilot study

Véronique Gille[1]*, Flore Gubert[1], Camille Saint-Macary[1], Stéphanie Dos Santos[2], Franck Houffoué[3], Hugues Kouadio[3], Epiphane Marahoua[3], Petanki Soro[4], Alexander van Geen[5]

1 IRD, UMR LEDa, PSL, Université Paris-Dauphine, Paris, France, 2 LPED (IRD/AMU), Centre Saint-Charles, Marseille, France, 3 Institute of Statistics and Applied Economics, Abidjan, Ivory Coast, 4 UFR of Earth Sciences and Mineral Resources, University Felix Houphouet Boigny, Abidjan, Ivory Coast, 5 Lamont-Doherty Earth Observatory, Columbia University, Palisades, New York, United States of America

* veronique.gille@ird.fr

## Abstract

Lead (Pb) exposure is a major global health concern, particularly for young children, yet awareness of the risks is low. Pb-based paint remains a significant source of exposure in many low- and middle-income countries, despite existing regulations. We investigate whether personalized information on lead in paint can increase awareness and encourage preventive behaviors. As part of a pilot study in Abidjan, Ivory Coast, painted surfaces in pregnant women's homes were tested using a low-cost Pb detection kit, followed by confirmatory testing with an X-ray fluorescence (XRF) device. Among the final sample of 153 women, those living in homes that tested positive for Pb were 33-35 percentage points more likely to acknowledge their exposure risk. This increased awareness led to self-reported behavioral changes among mothers of young children, including a higher likelihood of preventing children from ingesting paint chips and washing their hands more frequently. We find no impact on self-reported home-cleaning or renovation behaviors. Our findings highlight the potential of personalized information to drive behavioral change in environmental health.

## Introduction

Lead (Pb) exposure poses major health risks. According to the World Health Organisation, it caused 900,000 deaths and 21.7 million disability-adjusted life years in 2019 [1]. The global cost of Pb exposure was estimated at US$6 trillion — 6.9% of global GDP [2]. Pre-natal and early childhood exposure is particularly harmful, as it disrupts brain development [3] and is linked to impaired neurodevelopment [4], reduced cognitive abilities [5–7], and increased risk of juvenile incarceration [8]. Despite global efforts like banning leaded gasoline [9], many children, especially in low- and middle-income countries, remain highly exposed. It is estimated that 632

**Data availability statement:** All data files will be available on Datasuds (https://dataverse.ird.fr/) under DOI https://doi.org/10.23708/ZNQEVU after acceptance of the manuscript for publication.

**Funding:** Data collection for this study was financed by The Fund for Innovation in Development https://fundinnovation.dev/en. The funders had no role in study design, data collection and analysis, decision to publish, or preparation of the manuscript.

**Competing interests:** The authors have declared that no competing interests exist.

million children have blood lead levels exceeding the U.S. Centers for Disease Control and Prevention (CDC) previous reference level of 5 micrograms per deciliter ($\mu$g/dL), which was in place at the time of the cited study [10]; the CDC has since lowered this value to 3.5 µg/dL, so this figure likely underestimates the true global burden.

This paper focuses on lead in paint, a significant and preventable source of exposure in children [11,12]. Over half of countries still lack binding regulations on Pb in paint, including many in Sub-Saharan Africa. In Ivory Coast, where our study takes place, Pb-based paints remain common despite legal bans. A 2017 study showed that 63% of solvent-based paints in Abidjan contained Pb-levels at or above the most stringent limit of 90 ppm adopted by many countries [13]. In the assessment we conducted in 2023, 14 out of the 23 tested paint samples were above 500 ppm.

Given the heightened vulnerability of young children to lead exposure and toxicity, pregnant women and mothers of young children, as primary caregivers, represent the population group that should be most thoroughly informed. Exposure to Pb-based paint can be reduced through simple measures (dust cleaning, toy washing, covering paint), yet low awareness, especially among pregnant women and mothers of young children, remains a major barrier. Among 200 pregnant women in our Abidjan sample, only 19% knew Pb-exposure poses a risk to human health at baseline, and a mere 3% identified paint as a potential source of Pb-exposure. This mirrors findings in other countries [14,15], and underscores a global awareness gap.

This paper explores whether awareness of Pb exposure can be raised and whether information diffusion translates into the adoption of preventive measures. We focus on the role of *personalized* information about presence of Pb-based paint at home. While generic communication, whether through mass media or door-to-door information campaigns, has been shown to efficiently increase knowledge on health related risks [16], it is rarely sufficient to change behaviors [17,18]. Beliefs about one's vulnerability to a specific health risk are key to the take-up of preventive measures [19,20]. Delivering personalized information about Pb exposure risks is challenging and costly, as it requires the use of X-ray fluorescence analyzers (XRF). These devices are expensive ($\approx USD 30,000$) and can only be operated by trained professionals. Our study mobilizes an innovation: a low cost ($< \$1$ per test), easy-to-use kit that allows testing for the presence of Pb-based paint in individual homes. The sensitivity and specificity of the kit were recently shown to be on the order of 95% relative to a threshold of 500 mg/kg Pb in dry paint [21]. This threshold corresponds to one-tenth of the concentration of 5,000 mg/kg (0.5%) used by the US Environmental Protection Agency to define Pb-based paint but six times higher than the maximum Pb concentration of 90 mg/kg allowed in paint sold in the US today and in many other countries.

Our pilot intervention was conducted in Abidjan, Ivory Coast, on a sample of 200 pregnant women. Each participant was visited three times and received both generic information on the risks associated with Pb exposure and personalized information on the presence of Pb-based paints in their homes. In each household, paint testing was conducted twice by our team: first, through the use of demonstration kits applied

to two distinct painted surfaces during the initial visit; and second, via an X-ray fluorescence (XRF) test conducted during the second or third visit. Each participating woman received a kit free of charge at the end of the first visit and was also encouraged to utilize it independently on other painted surfaces. We evaluate the impact of personalized information on Pb exposure, by looking at the effect of the kit and XRF results on two sets of outcomes at endline (third visit): first, awareness on Pb exposure, and second, the (self-declared) adoption of preventive measures. We argue that our identification strategy yields causal estimates, given that the presence of Pb-based paint is not correlated with household characteristics.

Our first result is that personalized information is successful at improving awareness of Pb exposure risks. During the third visit, the share of women who mentioned that they were exposed to lead was 34 percentage points higher in the group where the kit or the XRF detected Pb in paint than in the group where no lead was found. We also find evidence that this enhanced awareness translates into the adoption of preventive measures among women with young children: those living in homes where both the demonstration kit and the XRF detected paint were 23 percent more likely to declare that they were preventing their children from eating paint chips, and 41 percent more likely to declare that they were washing their children's hands more often. By contrast, we find no impact on cleaning or renovation behaviors, likely due to the already high baseline frequency of cleaning and the longer-term, costly nature of renovations.

These results highlight a promising new approach to addressing Pb exposure by demonstrating that personalized information about Pb exposure risks can improve awareness and encourage the adoption of preventive behaviors. Our findings contribute to the limited body of literature examining the impact of personalized information in the context of other environmental pollutants, which has shown that such tailored communication is effective in altering beliefs about individual vulnerability and prompting preventive actions [17,18,22]. In the specific context of lead, most educational interventions, primarily conducted in the U.S., have so far been unsuccessful in reducing Pb exposure among children [23,24]. Two notable exceptions achieved success in enhancing knowledge and driving behavioral changes through group-based approaches in Bangladesh and the U.S [25,26]. More research is needed to assess whether the behavioral changes we observe after providing personalized information are sustained in the long term, beyond the immediate post-intervention period. In addition, future studies should validate these outcomes using objective measures, such as children's blood lead levels, rather than relying solely on self-reported behaviors. Such evidence would be critical to establish whether personalized information can translate into meaningful and lasting health benefits.

## Materials and methods

### Pilot intervention

Our pilot intervention on lead in paint was conducted between September 11, 2023, and March 3, 2024. The project protocol was approved by Columbia University IRB, under the number AAAU5543 (or IRB-AAAU5543) and the National Ethics Committee for Life and Health Sciences (CNESVS) from Ivory Coast, reference number 006- 23/MSHPCMU/CNESVS-km. The objective of this pilot was twofold: first, to diagnose the extent of Pb exposure risks in a city like Abidjan and, second, to raise parental awareness to prevent Pb toxicity in children and assess whether this leads to the adoption of preventive measures.

Our focus here is exclusively on the learning and behavioral aspects of our intervention. We recruited 200 pregnant women from 11 health clinics across the city of Abidjan at the time of their first prenatal consultation, of whom 153 were ultimately visited three times. Written informed consent was obtained in two stages. First, a household representative signed a paper form. Second, participants provided consent on a tablet, either by signing directly on the device or by confirming electronically. During the first visit, we collected baseline socio-demographic information on the respondents, their family and their dwelling. We also distributed leaflets delivering information on lead (health implications of Pb exposure, sources of Pb exposure and preventive measures to avoid contamination from Pb in paint, see S1 Fig), as well as an easy-to-use kit, to enable women to test the presence of Pb-based paint in their home. For demonstration purposes,

the surveyors tested the kit on two painted surfaces of each respondent's home. This is the first source of personalized information that we exploit in this paper.

During the second and third visits, approximately separated by one month, we asked questions related to knowledge about lead, awareness of Pb exposure and preventive measures. In addition, we tested for the presence of Pb on all painted surfaces of respondents' homes using a handheld X-ray fluorescence (XRF) analyzer. This is our second source of personalized information. The timing of the XRF was randomized, such that measures were made for 75 houses (48%) during the second visit, and 82 (52%) during the third visit.

Table 1 presents the information provided to participants prior to the third visit. Among the 153 women visited during the third wave, 69.9% (resp. 30.1%) were informed of the presence (absence) of Pb-based paint in their home. The randomization of the XRF timing introduced variation in how women received this information. Among those informed that lead was present, 39.2% received the information solely through the demonstration tests, primarily because they were randomized to receive the XRF assessment during the third visit. Another 10.4% were informed solely based on XRF results, either because lead was detected on surfaces not tested during the demonstration or because the XRF identified lead in underlying paint layers [21]. Finally, 20.3% were informed based on both the demonstration test and XRF.

Lastly, some women may have obtained information about the presence of Pb-Based paint by using the testing kit left with participants for self-use after wave 1. About 55% of the respondents reported using the kit in wave 3, and 92.9% of the self-reported results aligned with findings from either the demonstration test or the XRF.

**Empirical specification.** Personalized information on the presence of Pb-based paint in women's homes was systematically delivered in wave 1 based on the demonstration tests and in wave 2 or 3 based on the XRF results. To assess the impact of this personalized information, we estimate the following specification:

$$Y_{ij} = \alpha_1 + \alpha_2 \text{PbPos}_{ij} + X_{ij}\alpha_3 + \gamma_j + \epsilon_{ij} \tag{1}$$

where $Y_{ij}$ is awareness about lead exposure by woman $i$ interviewed by surveyor $j$ in wave 3, PbPos is a dummy equals to 1 if the kit and/or the XRF detected Pb in at least one of the home's painted surfaces during waves 1 or 2, $X_{ij}$ is a vector of household characteristics. We worked with four surveyors who were responsible for recruiting participants, distributing flyers, conducting interviews across the three waves of surveys, and demonstrating the kit. $\gamma_j$ are surveyor dummies that account for variations in interaction and communication styles among surveyors, as well as differences across health centers, since each surveyor was assigned to one or two centers.

The identification of $\alpha_2$ relies on the assumption that the presence of Pb-based paint is orthogonal to characteristics that would also be correlated with awareness of Pb exposure and the adoption of preventive measures. While we cannot formally test this assumption, we show in the next section that there is no significant differences in observable characteristics between women who were informed that they had Pb-based paint and women who were informed that they had no Pb-based paint. This absence of correlation between the information on paint Pb content and observable characteristics provides some support to our identifying assumption.

**Table 1**. Information received about lead in paint before third visit.

| Information content | Percentage of Women |
|---|---|
| **Lead in Paint** | **69.9** |
| Demonstration Test Only | 39.2 |
| XRF Only | 10.4 |
| Both Methods | 20.3 |
| **No Lead in Paint** | **30.1** |

Note: Sample of 153 women visited during the third wave (endline).

We also estimate an alternative specification that differentiates between the sources of information about lead content in paint: the kit only, the XRF only or both.

$$Y_{ij} = \beta_1 + \beta_2 \text{PbPosKit}_{ij} + \beta_3 \text{PbPosXRF}_{ij} + \beta_4 \text{PbPosBoth}_{ij} + X_{ij}\beta_5 + \gamma_j + \epsilon_{ij} \tag{2}$$

## Results and discussion

### Descriptive statistics and balance check at baseline

Our survey data were analyzed using Stata 17. Table 2 reports the baseline socio-demographic characteristics of the women in our sample, along with key attributes of their households and homes. 55% were in their first trimester of

**Table 2**. Descriptive statistics at baseline and differences between groups with Pb-positive and Pb-negative information.

| | Mean | Std err. | Info about lead | |
| | | | Diff | Std err. |
| | | | (Pos. − Neg.) | |
| | (1) | (2) | (3) | (4) |
| **Women characteristics** | | | | |
| First trimester pregnancy | 0.55 | 0.50 | 0.10 | 0.09 |
| Has a partner | 0.84 | 0.37 | 0.06 | 0.06 |
| Age | 28.00 | 6.07 | 1.25 | 1.08 |
| No education | 0.29 | 0.46 | -0.01 | 0.08 |
| Primary education | 0.24 | 0.43 | -0.03 | 0.08 |
| Secondary education | 0.29 | 0.46 | -0.06 | 0.08 |
| Tertiary education | 0.17 | 0.37 | 0.09 | 0.06 |
| Salaried | 0.13 | 0.34 | 0.03 | 0.06 |
| Self-Employed | 0.44 | 0.50 | 0.10 | 0.09 |
| No work | 0.43 | 0.50 | -0.14 | 0.09 |
| **House characteristics** | | | | |
| Nb. rooms | 2.69 | 1.37 | 0.04 | 0.23 |
| Nb. painted surfaces | 2.92 | 1.11 | 0.46** | 0.19 |
| **Household characteristics** | | | | |
| Household size | 3.75 | 1.55 | 0.57** | 0.27 |
| Nb. children < 5 yrs | 0.54 | 0.63 | 0.17 | 0.11 |
| Wealth score | 0.00 | 1.76 | 0.32 | 0.31 |
| House owner | 0.08 | 0.27 | 0.01 | 0.05 |
| **Pb knowledge** | | | | |
| Pb exposure is dangerous | 0.31 | 0.46 | 0.06 | 0.08 |
| Paint is a source of Pb exposure | 0.03 | 0.16 | 0.04 | 0.03 |
| Exposed to Pb | 0.17 | 0.37 | 0.06 | 0.07 |
| **Behavior** | | | | |
| Clean house at least twice per week | 0.90 | 0.31 | 0.04 | 0.05 |
| Use mop at least twice per week | 0.85 | 0.36 | 0.02 | 0.07 |
| Washed toys in last two weeks | 0.28 | 0.45 | -0.22 | 0.13 |
| Wash kids' hands bf. eat (oft./alw.) | 0.82 | 0.38 | -0.16 | 0.12 |
| Wash kids' hands bf. sleep (oft./alw.) | 0.12 | 0.33 | 0.03 | 0.09 |
| Wash kids' hands af. outside (oft./alw.) | 0.30 | 0.46 | 0.10 | 0.14 |
| Nb. observations | 200 | | 153 | |

* $p < 0.1$, ** $p < 0.05$, *** $p < 0.01$. All variables are measured at baseline. Columns (1) and (2) report the full sample means and standard errors. Columns (3) and (4) report the differences and associated standard errors in characteristics between between women with Pb-positive ("Positive") and Pb-negative ("Negative") information for the sample of women visited three times. The wealth score is a composite index constructed using Principal Component Analysis (PCA) on household assets and amenities.

pregnancy, most had a partner (84%) and the average age was 28. The sample reflects diverse socio-economic backgrounds, with variations in education, occupation, and housing conditions. Knowledge about Pb exposure was limited. While 31% stated that Pb exposure was dangerous, only 3% identified paint as a potential source of Pb exposure, and 17% believed they were exposed to lead in their daily lives. We also gathered data on hygiene practices. Most women reported frequent house cleaning —90% cleaned their homes more than twice a week, and 85% included mopping. Yet, among those with children under five, only 28% had washed their children's toys in the past two weeks. Handwashing practices were also not systematic. While 82% of respondents reported washing their children's hands before meals, only 12% did so before bedtime, and 30% after the children returned from playing outside.

Columns (3) and (4) report the differences and associated standard error in main characteristics at baseline between respondents with Pb-positive and Pb-negative information for the sample of women visited during wave 3. No significant differences are observed across the two groups, except with respect to household size and number of painted surfaces. This is evidence that the presence of Pb-based paint at home is not correlated with observable household characteristics, and that our empirical specification is likely to yield a causal estimate of the effect of personalized information.

Qualitative reports from our surveyors indicate that a key reason for attrition was that several women returned to their parents' homes to give birth. S1 Table examines whether the attrition of 47 women between waves 1 and 3 is selective with respect to observable characteristics. We find that attriters are more likely to have a partner, come from smaller households, and are more likely to have tertiary education. Reassuringly, the probability of remaining in the sample is not associated with the lead-in-paint information provided during the demonstration test in wave 1, with knowledge about lead, or with cleaning and hygiene behaviors.

### Effect of information on knowledge about lead and Pb-exposure

**Effect of generic information on overall knowledge about lead.** During our first visit, leaflets were distributed to all the participating women to inform them on Pb risks, sources of Pb exposure and preventive measures (see S1 Fig). Figs 1 and 2 illustrate the evolution of women's beliefs and knowledge about lead between waves 1 and 3. Specifically, Fig 1 highlights changes in women's understanding of the health consequences of Pb exposure, while Fig 2 focuses on their knowledge of Pb exposure sources. Two key findings emerge. First, this generic information has resulted in a significant change in beliefs about the health risks associated with Pb exposure. In wave 3, nearly 100% of respondents stated that Pb exposure was harmful to health, detrimental to a baby during pregnancy, and harmful during child growth (Fig 1). This contrasts sharply with wave 1, when approximately 75% of respondents reported being unaware of these risks. Second, knowledge of Pb exposure sources has similarly increased. In wave 3, nearly all respondents identified paint as a source of Pb exposure, compared to approximately 75% in wave 1, who reported knowing no source. Additionally, a substantial proportion of respondents identified water, cosmetics and metal cooking utensils as other potential sources of Pb exposure.

Fig 3 shows the evolution of women's knowledge on preventive measures between waves 1 and 3. In response to the open-ended question, "What measures do you think can be taken to prevent children's exposure to lead?", most women reported having no idea at baseline, and only a few mentioned children's hand-washing, home cleaning, removing paint chips, or repainting. In wave 3, responses to the same open-ended question suggest a significant improvement in women's knowledge, as nearly all of them were able to identify at least one preventive measure. The most cited ones were children's hand-washing and home cleaning, followed by preventing children from putting objects in their mouth, repainting and removing paint chips.

**Effect of personalized information on knowledge about home's Pb-profile and Pb-exposure.** Table 3 presents the effects of personalized information estimated following Eqs (1) and (2) on two distinct measures of awareness: whether women believe they are exposed to Pb in their daily lives and whether they think there is Pb-based paint in their

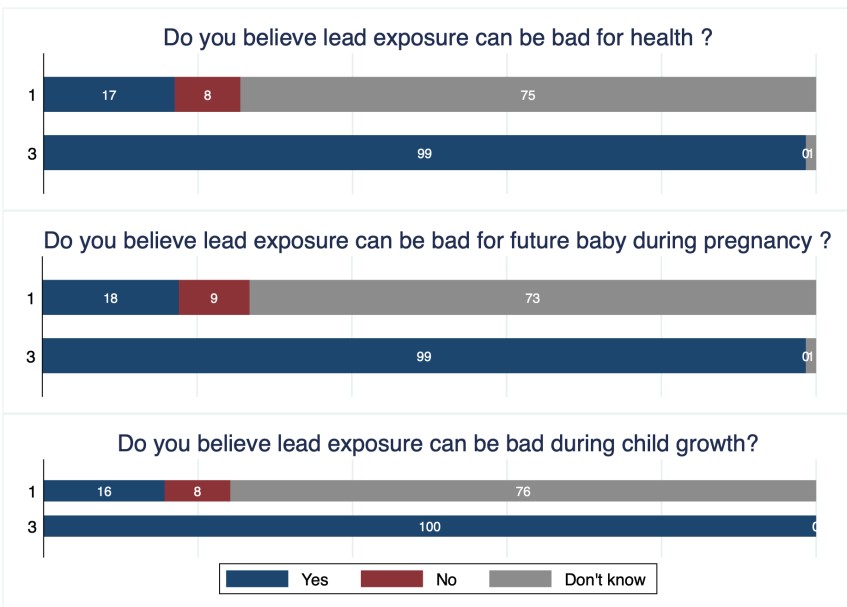

**Fig 1**. **Evolution of beliefs on health consequences of Pb exposure.**

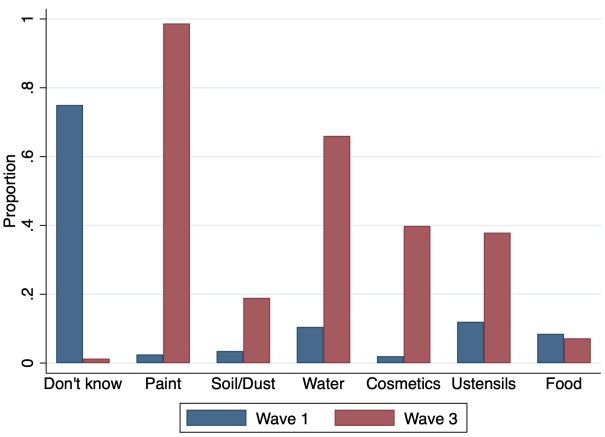

**Fig 2**. **Evolution of knowledge on sources of Pb exposure.**

home. Personalized information about the presence of Pb-based paint is found to significantly influence women's awareness of Pb exposure risks across both measures. Overall, women are 33-35 percentage points more likely to report that they are exposed to Pb or aware of its presence when the kit and/or the XRF detected Pb in at least one of their home's painted surfaces (columns 1 and 4). The effect is strongest when the information came from both tests, although the coefficients are generally not statistically different from each other (columns 2 and 5). The exception is column 5, where the coefficient on information from both tests is statistically different from the coefficient on information from XRF. Columns 3 and 6 test the robustness of the results when including information obtained through self-use of the kit. We use this specification solely as a robustness check, since the estimated coefficient may be biased: women received information from the self-test only if they opted to use it —a choice that could be influenced by unobserved factors also related to the outcome.

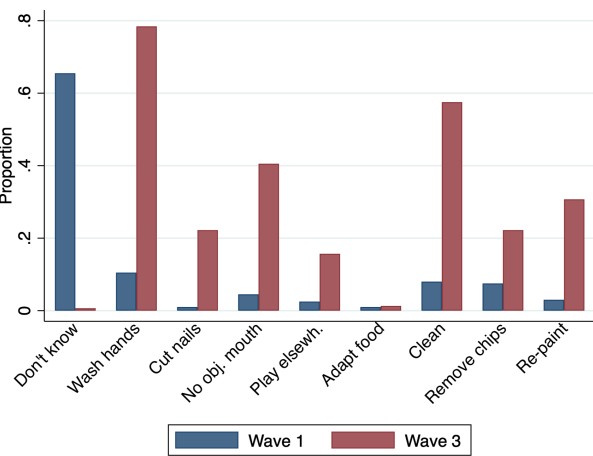

**Fig 3. Evolution of knowledge on preventive measures.**

**Table 3. Impact of Pb results on awareness of Pb exposure risks.**

| | Do you think that you are exposed to lead in your daily life? | | | I know there is lead | | |
|---|---|---|---|---|---|---|
| | Outcome: Yes | | | Outcome: Yes | | |
| | (1) | (2) | (3) | (4) | (5) | (6) |
| Lead detected | 0.333*** (0.0673) | | | 0.345*** (0.0799) | | |
| Lead: Demonstration test only | | 0.319*** (0.0743) | 0.250*** (0.0741) | | 0.341*** (0.0872) | 0.262*** (0.0872) |
| Lead: XRF only | | 0.265** (0.110) | 0.245** (0.106) | | 0.167 (0.130) | 0.144 (0.125) |
| Lead: XRF & Demonstration test | | 0.409*** (0.0935) | 0.350*** (0.0913) | | 0.462*** (0.110) | 0.396*** (0.107) |
| Lead: Self-test | | | 0.240*** (0.0701) | | | 0.274*** (0.0824) |
| Demographic controls | Yes | Yes | Yes | Yes | Yes | Yes |
| Surveyor dummy | Yes | Yes | Yes | Yes | Yes | Yes |
| Outcome mean (No Lead detected) | 0.57 | 0.57 | 0.57 | 0.22 | 0.22 | 0.22 |
| Observations | 153 | 153 | 153 | 153 | 153 | 153 |

* $p < 0.1$, ** $p < 0.05$, *** $p < 0.01$. The outcome variables are measured at endline (wave 3). Lead detected indicates whether lead was detected in the house before wave 3, either from the XRF or demonstration tests. Demographic controls include months into pregnancy, respondent's education level and age, education level of the household head, household size, number of children under 5 years old in the household, the household's wealth score, ownership or leasing status of the house, and the number of painted surfaces with different paints in the house. The wealth score is a composite index constructed using Principal Component Analysis on a set of household assets and amenities. Coefficients for covariates are reported in S2 Table.

Moreover, this information is self-reported and therefore subject to error and bias. While the coefficient on this source of information is large and statistically significant, the coefficients on other sources of information are only marginally affected.

Since the question "Do you think that you are exposed to Pb in your daily life" was asked at each wave, we can test the robustness of our findings using a panel specification with individual fixed effects. S3 Table confirms that the increase in awareness of Pb exposure risks is not driven by unobserved individual characteristics.

## Effect of personalized information on preventive behavior

Does the additional knowledge women acquire about their Pb exposure risks translate into the adoption of preventive measures? The preventive actions communicated to women via the leaflet distributed during our initial visit fall into three categories: cleaning measures, renovation measures, and child-specific measures. As many women in our sample were already frequently cleaning their homes at baseline (see Table 2), the scope for improvement of the first category of measures was very limited. The same holds true for renovation as it involves significant costs in terms of both time and money, making it more likely to occur over a longer time frame. We therefore focus primarily on measures concerning children.

Results are shown in Table 4. They rely on the same specification as before, estimated on the subsample of women with children under the age of five. For brevity, we report only the specification that differentiates the sources of personalized information. While there is no significant change in the probability of keeping play areas clean and regularly washing bottles and toys (col. 2), of cutting children nails more regularly (col. 5), or in the location where children play (col. 3), we observe a substantial effect of personalized information on two other measures: women informed that they live in homes that tested Pb-positive according to both the kit and the XRF were 23 percentage more likely to report preventing their children from ingesting paint chips (col. 1), and 41 percentage points more likely to report increasing the frequency of children's handwashing (col. 4). Women solely informed by the kit results also reported a higher likelihood of preventing paint-chip ingestion. Given the large number of outcomes being considered, we also report sharpened q-values (in square brackets) [27], which account for multiple hypothesis testing. We use the two-stage adaptive Benjamini procedure, as it provides substantially greater power than family-wise error rate methods such as Bonferroni or Holm. The effect remains only significant for children's hand-washing frequency. A larger sample size would be necessary to reach a definitive conclusion regarding prevention of paint-chip ingestion.

Table 5 presents the effects of personalized information on the other types of preventive measures (cleaning and renovating).

**Table 4**. Impact of Pb results on take-up of child-related preventive measures.

| | Did you take any preventive measures? | | | | |
| --- | --- | --- | --- | --- | --- |
| | Child related measures | | | | |
| | No chip mouth (1) | Clean play (2) | Chge play (3) | Wash hands (4) | Cut Nails (5) |
| Lead: Demonstration test only | 0.170* | -0.075 | -0.003 | 0.075 | 0.059 |
| | (0.09) | (0.09) | (0.09) | (0.12) | (0.09) |
| | [0.48] | [1.00] | [1.00] | [1.00] | [1.00] |
| Lead: XRF only | -0.044 | -0.016 | -0.152 | 0.118 | -0.060 |
| | (0.11) | (0.11) | (0.11) | (0.14) | (0.11) |
| | [1.00] | [1.00] | [1.00] | [1.00] | [1.00] |
| Lead: XRF & Demonstration test | 0.225** | -0.045 | -0.082 | 0.406*** | -0.031 |
| | (0.10) | (0.10) | (0.09) | (0.12) | (0.10) |
| | [0.20] | [1.00] | [1.00] | [0.02] | [1.00] |
| Demographic controls Surveyor dummy | Yes Yes | Yes Yes | Yes Yes | Yes Yes | Yes Yes |
| Mean for Kit result: no lead Observations | 0.00 77 | 0.05 77 | 0.05 77 | 0.05 77 | 0.00 77 |

\* $p < 0.1$, \*\* $p < 0.05$, \*\*\* $p < 0.01$. The outcome variables are measured at endline (wave 3). Demographic controls include months into pregnancy, respondent's education level and age, education level of the household head, household size, number of children under 5 years old in the household, the household's wealth score, ownership or leasing status of the house, and the number of painted surfaces with different paints in the house. The wealth score is a composite index constructed using Principal Component Analysis on a set of household assets and amenities. Sharpened q-vals are reported in square brackets [27]. Coefficients for covariates are reported in S4 Table.

**Table 5**. Impact of test results on take-up of preventive measures - cleaning and renovating.

| | Did you take any preventive measures? | | | | |
| --- | --- | --- | --- | --- | --- |
| | Cleaning | | | Renovate | |
| | Clean surfaces (1) | Clean doors & windows (2) | Remove paint chips (3) | Renovate (4) | Careful during renovation (5) |
| Lead: Demonstration test only | -0.038 | -0.041 | -0.063 | -0.058 | 0.064 |
| | (0.06) | (0.07) | (0.05) | (0.05) | (0.04) |
| | [1.00] | [1.00] | [1.00] | [1.00] | [1.00] |
| Lead: XRF only | -0.029 | -0.078 | -0.131* | -0.087 | 0.006 |
| | (0.09) | (0.11) | (0.08) | (0.07) | (0.06) |
| | [1.00] | [1.00] | [1.00] | [1.00] | [1.00] |
| Lead: XRF & Demonstration test | 0.022 | -0.074 | -0.069 | -0.011 | 0.040 |
| | (0.08) | (0.09) | (0.07) | (0.06) | (0.05) |
| | [1.00] | [1.00] | [1.00] | [1.00] | [1.00] |
| Demographic controls | Yes | Yes | Yes | Yes | Yes |
| Surveyor dummy | Yes | Yes | Yes | Yes | Yes |
| Mean for Kit result: no lead | 0.07 | 0.35 | 0.11 | 0.04 | 0.00 |
| Observations | 153 | 153 | 153 | 153 | 153 |

* $p < 0.1$, ** $p < 0.05$, *** $p < 0.01$. The outcome variables are measured at endline (wave 3). Demographic controls include months into pregnancy, respondent's education level and age, education level of the household head, household size, number of children under 5 years old in the household, the household's wealth score, ownership or leasing status of the house, and the number of painted surfaces with different paints in the house. The wealth score is a composite index constructed using Principal Component Analysis on a set of household assets and amenities. Sharpened q-vals are reported in square brackets [27]. Coefficients for covariates are reported in S5 Table.

As expected, Table 5 shows that personalized information about the presence of Pb-based paint does not lead to any increase in the adoption of either cleaning or renovation measures.

## Conclusion

Beliefs about one's vulnerability to health risks are one of the barriers to adopting preventive measures, and information plays a crucial role in shaping these beliefs. In the case of Pb exposure, especially from Pb-based paint, information about exposure risks is costly to obtain because it requires expert intervention. This paper evaluates whether an innovation — a low-cost, easy-to-use kit — can help mitigate Pb exposure risks by providing personalized information on Pb exposure. We find that providing generic information on Pb exposure risks, along with personalized information on the presence of Pb-based paint, raises households' awareness about Pb and enhance the adoption of preventive measures that can effectively reduce children's exposure. Yet, we do not find impacts on home cleaning and renovation behaviors, which is likely due to the already high frequency of home cleaning declared at baseline, and to the longer-term and costly nature of renovations.

Our analyses are subject to certain limitations that we plan to address in further research. First, they rely on a small number of observations, reflecting the pilot nature of this study. The external validity of our results is thus limited. Second, all outcomes are self-reported and may be affected by social desirability bias. Third, our estimates capture only short term impacts, and we do not know how they persist over time and whether mothers' behavioral changes translate into a decrease in children's blood lead levels. We intend to scale up this pilot study into a larger experiment in Abidjan to address these limitations. We will also investigate which strategies are most effective for disseminating generic information on Pb hazards and delivering personalized home Pb profiles at reasonable costs, with the objective of informing policy recommendations.

## Supporting information

**S1 Fig. Leaflet on Pb risks, sources of Pb exposure and preventive measures.**
(PDF)

**S1 Table. Attrition analysis.**
(PDF)

**S2 Table. Estimated coefficients for covariates in Table 3.**
(PDF)

**S3 Table. Robustness check with panel specification.** Impact of tests' results on awareness of lead exposure risks: panel specification.
(PDF)

**S4 Table. Estimated coefficients for covariates in Table 4.**
(PDF)

**S5 Table. Estimated coefficients for covariates in Table 5.**
(PDF)

## Acknowledgments

We thank Elise Huillery, Björn Nilsson and Elsa Perdrix for their useful comments. Language editing was assisted by ChatGPT (OpenAI). The authors reviewed and approved all content to ensure scientific accuracy and accept full responsibility for the manuscript.

## Author contributions

**Conceptualization:** Véronique Gille, Flore Gubert, Camille Saint-Macary, Stéphanie Dos Santos, Franck Houffoué, Hugues Kouadio, Alexander van Geen.

**Data curation:** Véronique Gille, Flore Gubert, Camille Saint-Macary, Franck Houffoué, Epiphane Marahoua, Petanki Soro, Alexander van Geen.

**Formal analysis:** Véronique Gille, Flore Gubert, Camille Saint-Macary, Franck Houffoué, Alexander van Geen.

**Funding acquisition:** Véronique Gille, Flore Gubert, Camille Saint-Macary, Stéphanie Dos Santos, Hugues Kouadio, Alexander van Geen.

**Methodology:** Véronique Gille, Flore Gubert, Camille Saint-Macary, Stéphanie Dos Santos, Franck Houffoué, Hugues Kouadio, Petanki Soro, Alexander van Geen.

**Project administration:** Flore Gubert, Hugues Kouadio, Alexander van Geen.

**Supervision:** Flore Gubert, Hugues Kouadio, Alexander van Geen.

**Visualization:** Véronique Gille, Flore Gubert, Camille Saint-Macary, Franck Houffoué, Petanki Soro, Alexander van Geen.

**Writing – original draft:** Véronique Gille, Flore Gubert, Camille Saint-Macary, Alexander van Geen.

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
