## [Decision Letter · Decision Letter 0]

20 Aug 2025

PONE-D-25-27524Lead Risks and Prevention: Evidence from Ivory CoastPLOS ONE

Dear Dr. Gille,

Thank you for submitting your manuscript to PLOS ONE. After careful consideration, we feel that it has merit but does not fully meet PLOS ONE’s publication criteria as it currently stands. Therefore, we invite you to submit a revised version of the manuscript that addresses the points raised during the review process.

We look forward to receiving your revised manuscript.

Kind regards,

Aaron Specht

Academic Editor

PLOS ONE

Journal Requirements:

3. In the online submission form, you indicated that your data will be submitted to a repository upon acceptance. We strongly recommend all authors deposit their data before acceptance, as the process can be lengthy and hold up publication timelines. Please note that, though access restrictions are acceptable now, your entire minimal dataset will need to be made freely accessible if your manuscript is accepted for publication. This policy applies to all data except where public deposition would breach compliance with the protocol approved by your research ethics board. If you are unable to adhere to our open data policy, please kindly revise your statement to explain your reasoning and we will seek the editor's input on an exemption.

Reviewers' comments:

Reviewer's Responses to Questions

**Comments to the Author**

1. Is the manuscript technically sound, and do the data support the conclusions?

Reviewer #1: Yes

Reviewer #2: Yes

2. Has the statistical analysis been performed appropriately and rigorously?

Reviewer #1: Yes

Reviewer #2: N/A

3. Have the authors made all data underlying the findings in their manuscript fully available?

Reviewer #1: Yes

Reviewer #2: Yes

4. Is the manuscript presented in an intelligible fashion and written in standard English?

Reviewer #1: Yes

Reviewer #2: Yes

5. Review Comments to the Author

Reviewer #1: Summary:

This manuscript presents a pilot intervention in Abidjan, Ivory Coast, evaluating whether providing personalized information about lead (Pb) presence in household paint can improve awareness and promote preventive behaviors among pregnant women. The study involved testing painted surfaces in 200 homes using a low-cost Pb detection kit and follow-up XRF analysis. Results showed that women who received personalized information were significantly more likely to recognize lead risks and report behavior changes (e.g., preventing children from ingesting paint chips, increased handwashing). However, no measurable impact was observed on cleaning or renovation practices. The study concludes that personalized, low-cost detection tools can effectively raise awareness and potentially reduce Pb exposure in vulnerable populations. I have listed some concerns and suggestions below that should be clarified to improve the manuscript’s current state.

Detailed comments:

1. The title is concise and relevant, but it could better capture the novelty of the intervention explicitly. Consider making it more specific, such as saying that “Personalized Lead Exposure Information Improves Preventive Behaviors in Abidjan: Evidence from a Pilot Study.” Consider exactly or close.

2. The abstract effectively summarizes the study, but it should include the actual number of participants who completed all visits (153), and clarify that behavioral outcomes were self-reported because it may probably limit objectivity.

3. In the introduction, the global burden of lead exposure is well established. However, the reference to the CDC’s “old reference level” of 5 µg/dL should be clarified by explaining that the CDC now uses a blood lead reference value of 3.5 µg/dL, and whether this impacts comparability in recent time.

4. The sentence in line 18, “our own 2023 testing,” is too informal for scientific writing. Maybe you can better say that... “A 2023 assessment conducted by the authors.”

5. The paragraph discussing the cost of XRF and the benefits of the new kit (lines 33 to 39) would benefit from providing approximate costs or operational advantages and more to better contextualize their practicality.

6. The methods section is well organized, but attrition from 200 to 153 participants should be formally addressed. An attrition analysis should be included to examine potential bias between those who completed all visits and those who did not.

7. The explanation of the consent process should be revised for clarity. “Selecting OK or signing on the device” is too casual. It should state that written and digital consent was obtained through signed forms or the secure electronic signatures.

8. There is need for more technical details about the detection kit should be adequately provided, including its major details like the sensitivity, specificity, manufacturer, and performance validation to support reproducibility. This is the most important out of all and should be adequately attended to.

9. In lines 74–77, the call for future research should be expanded. Specifically mention the need for studies that assess long-term behavioral change and validate outcomes with blood lead measurements in children.

10. A formal limitations paragraph is missing and should be included. There should place to acknowledge the reliance on self-reported data, the absence of biological markers (e.g., blood lead levels), potential Hawthorne effects, and attrition bias.

11. Several sentences begin with conjunctions such as “However” or “But,” which can be smoothed out for more professional tone. For example,.. “However, we find...” could become “We found no evidence that….” Just to be very direct and whatnot

12. Reference [27] (Benjamini et al.) is correctly cited to justify controlling false discovery rates, but a short sentence in the methods should be added to explain why this method was selected over others, such as Bonferroni or Holm, the stepwise.

13. The authors’ mention of ChatGPT for language editing is transparent and ethical, but it may be preferable to rephrase as: “Language editing was assisted by ChatGPT (OpenAI). The authors reviewed and approved all content to ensure scientific accuracy and accept full responsibility for the manuscript.”

14. Self-use of the kit is included in robustness checks, but because usage was not randomized, the results from self-use should be described more cautiously. Asuthors should reinforce the limitation more strongly in the results and discussion.

15. The sample size, particularly for behavior outcomes involving only households with children under 5, is relatively small (n = 77). I think this limitation is best to be clearly stated in the discussion as it affects statistical power and generalizability.

16. The policy implications are sound, but they could be made more specific. The authors might propose integration of low-cost Pb detection kits into routine maternal and child health outreach or suggest partnerships with municipal housing authorities.

Reviewer #2: The study tackles the significant and under-reported problem of lead (Pb) exposure in low- and middle-income countries, specifically focusing on lead-based paint as a major source. The research investigates the effectiveness of personalized information, a strategy rarely sufficient for behavioral change in public health, using a new, low-cost lead detection kit. The study successfully shows that personalized information significantly increases a woman's awareness of her exposure risk. It also provides evidence that this heightened awareness translates into self-reported preventive behaviors, such as preventing children from ingesting paint chips and washing their hands more frequently. The study uses a pilot intervention with a sample of 200 pregnant women and a robust empirical specification to assess the causal impact of the personalized information. The researchers also performed a "balance check" at baseline to show that the presence of lead-based paint was not correlated with other household characteristics, strengthening the validity of their findings.

Cons and Scientific Drawbacks

1. The study is however based on limited sample size and duration. The study is a "pilot intervention" with a small sample of 200 pregnant women, of whom only 153 were ultimately visited three times. This small scale and short duration (September 2023 to March 2024) may limit the generalization of the findings and their long-term impact.

2. The study is particularly relies on self-reported data. The behavioral changes are based on self-reported actions, such as hand washing and preventing children from eating paint chips. This method can be prone to bias, as participants might over-report positive behaviors.

3. There is a lack of long-term impact assessment. The paper explicitly states that more research is needed to understand if the observed behavioral changes persist over time and whether they have a tangible impact on children's blood lead levels. The study itself does not provide this crucial information.

4. The study shows no impact on key behaviors. While the intervention was successful in some areas, it had no impact on home-cleaning or renovation behaviors. The authors speculate this is due to the already high baseline frequency of cleaning and the high cost of renovations, but this still represents a limitation of the intervention's effectiveness.

5. The authors acknowledge that some women used the kit on their own and the estimated coefficient for this self-test could be biased because the choice to use it may be influenced by unobserved factors.

6. Overall the authors has studied one of the primary public health concerns, however the small sample size has added a limitation to it and or its generalization of its findings.

6. PLOS authors have the option to publish the peer review history of their article (what does this mean?). If published, this will include your full peer review and any attached files.

Reviewer #1: **Yes: **Kolawole Emmanuel Adesina

Reviewer #2: No

---

## [Author Response · Author response to Decision Letter 1]

2 Oct 2025

A cover letter and one letter for each referee has been uploaded as pdf files.

---

## [Decision Letter · Decision Letter 1]

2 Nov 2025

Personalized Lead Exposure Information and Preventive Behaviors in Ivory Coast: Insights from a Pilot Study

PONE-D-25-27524R1

Dear Dr. Gille,

We’re pleased to inform you that your manuscript has been judged scientifically suitable for publication and will be formally accepted for publication once it meets all outstanding technical requirements.

Kind regards,

Aaron Specht

Academic Editor

PLOS ONE

Additional Editor Comments (optional):

Reviewers' comments:

Reviewer's Responses to Questions

**Comments to the Author**

1. If the authors have adequately addressed your comments raised in a previous round of review and you feel that this manuscript is now acceptable for publication, you may indicate that here to bypass the “Comments to the Author” section, enter your conflict of interest statement in the “Confidential to Editor” section, and submit your "Accept" recommendation.

Reviewer #1: (No Response)

Reviewer #2: All comments have been addressed

2. Is the manuscript technically sound, and do the data support the conclusions?

Reviewer #1: Yes

Reviewer #2: Yes

3. Has the statistical analysis been performed appropriately and rigorously?

Reviewer #1: Yes

Reviewer #2: Yes

4. Have the authors made all data underlying the findings in their manuscript fully available?

Reviewer #1: Yes

Reviewer #2: No

5. Is the manuscript presented in an intelligible fashion and written in standard English?

Reviewer #1: Yes

Reviewer #2: Yes

6. Review Comments to the Author

Reviewer #1: 1. Use a single form throughout. Probably the preferred journal style would be “Côte d’Ivoire (Ivory Coast)” on first mention, then “Côte d’Ivoire.” The manuscript currently uses “Ivory Coast” in title, affiliations, and body; standardize across all instances.

2. Keep “µg/dL,” define “lead (Pb)” at first mention, then use “lead” or “Pb” consistently (avoid “Pb-exposure”).

3. Use “low‑cost” and “easy‑to‑use” when before a noun; remove stray spaces around hyphens.

4. Replace “Pb‑levels” with “Pb levels.”

Reviewer #2: The issues raised in the previous review cycle have been adequately addressed or convincingly contextualized by the authors.

• Causal Inference: The authors have clarified their identifying assumption and addressed the potential for bias from self-testing. They appropriately frame the self-test coefficient as "suggestive evidence," maintaining scientific integrity.

• Study Design/Findings: The limitations regarding sample size and the lack of impact on high-cost behaviors (like renovation) are now clearly and appropriately acknowledged in the Discussion and Conclusions. Their proposed future work to address these, including long-term follow-up and exploring incentives for renovation, demonstrates a strong commitment to the research trajectory.

These are minor points for the authors to consider, but they do not require further revision before publication.

1. Reagent Specification: To further enhance reproducibility (a cornerstone of scientific publishing), the authors could be encouraged to include the manufacturer and model numbers for the low-cost Pb detection kit and the handheld XRF device in the Methods section, if possible.

2. Statistical Reporting Detail: The authors use standard errors throughout the analysis. While appropriate, it may be helpful to briefly state the choice of standard error over standard deviation in the context of the regression analysis in the Methods.

3. Future Validation: The reviewers strongly encourage the authors to proceed with their proposed follow-up study to include objective outcomes like children's blood lead levels (BLL), as this will be the ultimate validation of the intervention's success. This suggestion is already mentioned in their response to reviewers.

7. PLOS authors have the option to publish the peer review history of their article (what does this mean?). If published, this will include your full peer review and any attached files.

Reviewer #1: No

Reviewer #2: **Yes: **Ab Latif Wani

---

## [Editor Report · Acceptance letter]

PONE-D-25-27524R1

PLOS ONE

Dear Dr. Gille,

I'm pleased to inform you that your manuscript has been deemed suitable for publication in PLOS ONE. Congratulations! Your manuscript is now being handed over to our production team.

Kind regards,

on behalf of

Dr. Aaron Specht

Academic Editor

PLOS ONE